# Cluster randomized controlled trial to assess the effectiveness of a package of community-based interventions on continuum of maternal and newborn healthcare in Sidama, Ethiopia: The SiMaNeH trial protocol

**Achamyelesh Gebretsadik**[1]*, **Yemisrach Shiferaw**[1], **Hirut Gemeda**[1], **Yaliso Yaya**[2]

**1** School of Public Health, Hawassa University, Hawassa, Ethiopia, **2** Faculty of Health and Social Sciences, Western Norway University of Applied Sciences, Bergen, Norway

* achamyelesh@hu.edu.et

## Abstract

### Background

Maternal and newborn mortality and morbidity remain high in low- and middle-income countries such as Ethiopia. Limited access and dropouts from essential continuum of care interventions are critical factors. In Ethiopia, about one in five completes the continuum of essential care through pregnancy, childbirth, and the postnatal period. Evidence is limited on whether packages of interventions involving key community health actors increase the proportion completing essential maternal and newborn healthcare continuum in rural Sidama regional, state, Ethiopia.

### Objective

This study aims to implement and evaluate the cumulative effectiveness of a package of community-based interventions designed to enhance involvement of key community health actors to improve the completion rate of continuum of maternal care and utilization of essential newborn care.

### Methods

Twenty rural kebeles (clusters) in Sidama Regional State, Ethiopia, are randomly allocated to intervention and control arms. A total of 2000 pregnant women, 1000 per arm, will be recruited between 20th and 26th week of gestation after intervention. Then the pregnant mothers and their newborn babies will be followed until six weeks postpartum between June 2024 and February 2025. In the intervention arm, mothers and newborns will receive targeted interventions at home and in their community designed to improve the completion rate of recommended maternal and newborn care. Control clusters will receive normal care from the state public health system. Primary outcomes will be the difference in the completion of continuum of essential maternal and rate of use of essential and emergency

**Data availability statement:** No datasets were generated or analyzed during the current study. In accordance with ethical restrictions, details for requesting access to the relevant data will be provided upon study completion.

**Funding:** The author(s) received no specific funding for this work.

**Competing interests:** No authors have no competing interest.

newborn care and referrals between intervention and control clusters. These outcomes include rates of antenatal care completion, facility deliveries with skilled care, completion of at least four postnatal care contacts, and the overall completion of all the way from first antenatal visit through the postnatal care. Newborn outcomes will be measured through essential newborn care utilization and emergency (danger sign) identification and referrals. Secondary outcomes will include the effect of the intervention on reducing neonatal mortality and stillbirths.

## Conclusion

This trial will implement and evaluate a package of community-based interventions within existing community healthcare infrastructure. The outcome may inform evidence-based community-based decisions to improve the continuum of essential maternal and newborn care.

## Trial registration

The trial is registered at Pan African Clinical Trial Registry: PACTR202402782261294

## Introduction

Global maternal and child mortality was remarkably reduced between 2000 and 2015 through efforts under Millennium Development Goals (MDGs). However, recent estimates by UN inter-agency highlight that the progress stagnated between 2016 and 2020 [1]. In 2020, there were over 300,000 maternal deaths, 2.4 neonatal deaths, and another 1.9 stillbirths [2,3]. Furthermore, neonatal mortality, the death of a newborn baby within the first month of life, shares nearly half (47%) of mortality of under-5 children shows slow progress.

Over 99 percent of maternal and newborn deaths and severe morbidities and disabilities from pregnancy and childbirth occur in low-income and middle-income countries. Unfortunately, sub-Saharan Africa has the highest neonatal mortality rate in the world with 27 per 1000 live births [4], also contributing close to half (42%) of global stillbirths, the death of babies after 28th weeks of gestation and before birth, in 2019 [5]. Poor investments [6] and limited access to quality maternal and newborn care in resource-limited settings are important factors because one-third of the neonatal deaths occur within the first day and three-forth in the first week after birth [3]. Estimates from modeling 75 high-burdened countries highlighted improved coverage of essential maternal and newborn care interventions could avert over two-third of neonatal deaths, one-third of stillbirths, and over half of maternal deaths every year [7] because 85% of newborn deaths are because of fetal asphyxia, preterm birth, intra-partum complications and neonatal infections [7]. Nevertheless, in resource-limited settings such high coverage of essential care is challenging to achieve, especially in rural communities.

Since the WHO's Alma Ata declaration in 1978 [8], community-based interventions, intended to bring essential healthcare services closer to these hard-to-reach areas have emerged as important alternatives to improve maternal and newborn healthcare. However, the strategies and its success are not uniform across communities and countries.

Ethiopia, like other low-income countries, struggles with limited access to the life-saving interventions for maternal and newborn health to the larger part of its population living in the rural areas, underscoring the need for innovative and sustainable strategies to close the gap in adverse pregnancy and childbirth outcomes.

One such innovative approach is the Ethiopian Health Extension Program (HEP), a flagship national program rolled out in 2003 by Ministry of Health of Ethiopia [9]. In each village of about 5,000 population, two women known as Health Extension Workers (HEWs) trained for one year in general health are placed in a satellite health post to provide health promotion and disease prevention services to residents. These women are permanent employee of Ethiopian government with monthly salary.

The Health Extension Program (HEP) has been instrumental in expanding access to primary healthcare services, demonstrating the feasibility and impact of deploying health extension workers (HEWs) in rural settings. From 2012, another voluntary network of about 30 women per village known as women development team (HDT) were introduced to support the works of HEWs, creating a strong network closer to households. Consequently, a systematic review by Yibeltal Assefa and colleagues showed that the HEP has significantly improved essential preventive and health promotion indicators [10].

Yet, concerning the skills of the HEWs to provide essential maternal and newborn care, a study in Ethiopia identified up to 88% of HEWs had poor knowledge of neonatal danger signs and expressed that they are not confident enough to attend deliveries [11]. This underscores the importance of an approach that improves their confidence, skills, and integration with health facilities.

Furthermore, a 2022 article by Tiruneh GT and colleagues demonstrated that overall, only one-fifth (20%) of women who initiated the first antenatal care completed the continuum of recommended essential care that combines completion of antenatal care, skilled birth attendance, and critical postpartum care [12].

One of the challenges in maternal and newborn care is the poorly organized health systems that lack a mechanism for continuum of care completion even once a mother has visited for her first antenatal care or subsequent care. As such, very low proportion of women make postnatal contact with competent health professionals. In 2022, WHO published guidelines for postnatal care and recommends at least four such contacts in the key time periods during the first 24 hours, 48–72 hours, one-two weeks and six weeks after birth [13]. Without successful completion of postnatal care during these key periods, it is difficult to succeed in improving the continuum of care.

The low rate of completion of essential continuum of care highlights the importance of a community-based maternal and newborn care system that supports the mother, the baby and family through key community health stakeholders such as skilled and motivated community health workers, trained and supervised community opinion leaders and women support groups. A systematic review synthesized evidence from community-based interventions to improve maternal and newborn health by Zohra S Lassi and colleagues in 2015 showed that interventions with strong community involvement showed significant positive results compared to other community-based intervention without community involvement [14]. Previous community-based intervention trials in low-resource settings have demonstrated the important benefits of community-based interventions [15–17]. However, many of these trials have been limited by a narrow focus of addressing specific factors, overlooking the complex and interconnected nature of the various community-based actors and stakeholders that influence healthcare utilization in rural communities.

Considering these challenges, this protocol outlines a comprehensive approach to conducting cluster randomized rural trials implementing a package of interventions that includes enhancing involvement of several community actors. The current intervention includes training and supervision and evaluating the effectiveness of these interventions in improving the completion of maternal continuum of care and newborn essential care.

Our hypothesis is that the proposed community-based intervention package intended to engage a broad spectrum of key actors, including community health workers, women's

support groups, influential community leaders, local social welfare scheme systems (known as IDIR in Ethiopia), and the linkages between community and health facility services has a potential to improve maternal and newborn healthcare continuum outcomes compared to control group receiving the routine public health services. Continuum of maternal and newborn care is a continuity of care during pregnancy, childbirth, and postpartum period with coordinated efforts from home through the health facility [18]. To achieve improved rate of completion of the continuum of care, we aim to address the complex barriers such as poor knowledge of the importance of initiation and continuation of care until completion of standard care, transport barriers, and family care related challenges when a mother or baby needs to travel to reach care.

## Overall aim

The aim of the proposed trial is to evaluate the collective effectiveness of a package of community-based interventions on improving the continuum of maternal and newborn healthcare.

## Objectives

1) To evaluate whether a set of community-based interventions improves completion of maternal continuum of care: A) The overall completion rate from the first antenatal visit through at least four postnatal care contracts and B) completion rates of each three components of the continuum of maternal care (antenatal, delivery in a health facility, and postnatal care completions). C) Explore the experience of mothers and care providers on the effect of the intervention for maternal care, including the challenges and opportunities.

2) To determine the effectiveness of the community-based strategies on improving utilization of essential and emergency newborn care: A) WHO recommended four elements of essential newborn (immediate and thorough baby drying, skin-to-skin contact, initiation of early breast feeding within first hour, and delayed cord clamping). B) Identification and referral of severely sick babies such as babies with danger signs. C) Cost-effectiveness of the intervention on improving maternal and newborn continuum of health care.

## Methods

### Setting

This trial research project will take place in rural areas of Sidama, a regional state in southern part of Ethiopia. Sidama became an autonomous regional state in 2020, separating from the former Southern Nations, Nationalities, and Peoples Regional State (SNNPRS), which is divided into four new regional states between 2020 and 2023. Sidama is situated approximately 400 kilometers south of Addis Ababa with an estimated population of about 4.6 million people in 2023. The capital of the regional state is Hawassa (See Fig 1). Still, more than 80 percent of the population lives in rural areas, where agriculture is the primary economic activity and access to quality healthcare services is limited.

In 2018, an empirical study in one of the districts in Sidama showed that fertility was lower than reported from national estimates with crude birth rate of 22.8 births per 1000 population and total fertility rate of 2.9 children per woman. Furthermore, the population is transitioning to a low-mortality and low-fertility rate [19]. However, the United Nations projected crude birth rate for Ethiopia is 30.6 births per 1000 population in 2024 [20], far higher than the empirical findings (Table 1, Fig 1).

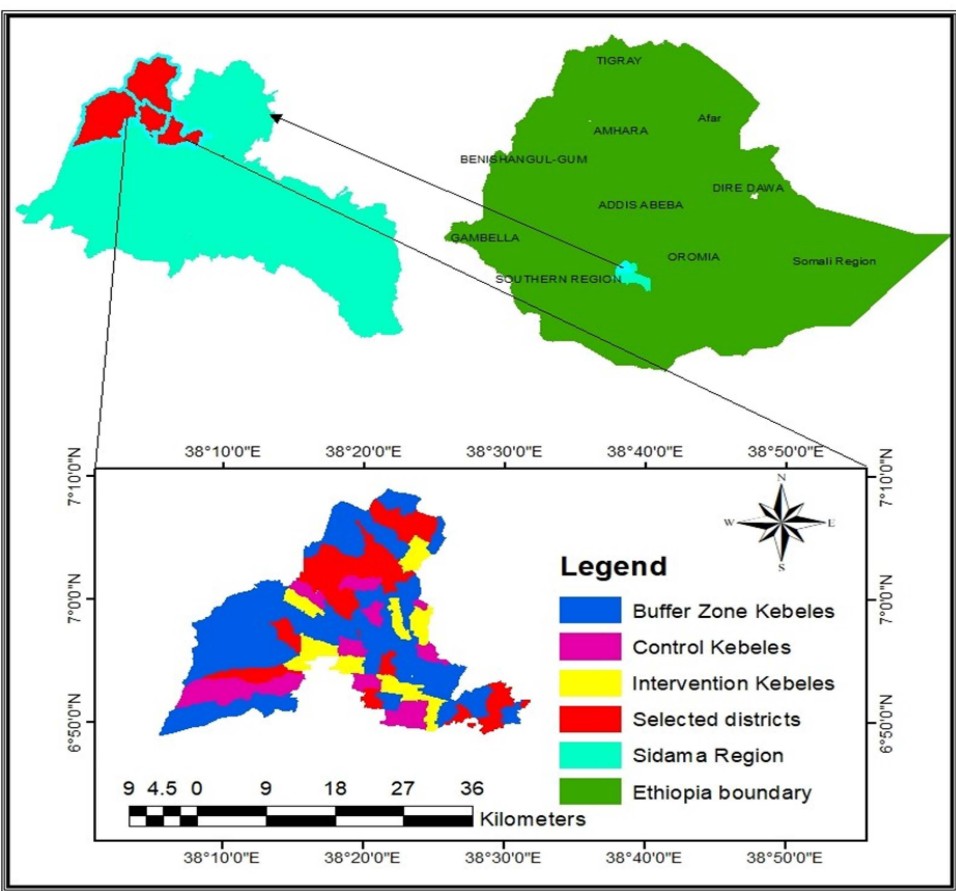

**Fig 1. The map of the study area within Sidama regional State, Ethiopia.** (Authors created this figure and give permission to the editor under CC BY 4.0 license).

**Table 1. Background demographic and health service characteristics of study population.**

| Background | Intervention | Control | Remarks |
|---|---|---|---|
| Total population in 2024 | 84 347 | 87 830 | |
| Expected births in 2024$^x$ | 2 530 | 2 634 | |
| Number of existing health centers | 7 | 8 | some overlaps |
| Number of Health Extension Workers | 37 | 35 | |
| Number of Health Developments Team | 545 | 560 | |
| Average distance to health centers | 5.3 km | 5,1 km | |

*Note: $^x$ = annual birth rate (3% of population) multiplied by population*

## Theoretical and conceptual frameworks

The trial will employ theoretical and conceptual frameworks, including the WHO-endorsed Participatory Learning and Action (PLA) model [21]. The PLA will be used as a guiding model to be used by a team of community resource persons comprised of religious, administrative, and local social scheme leaders, and women support groups. The resource (support) group will use the model to identify, prioritize, address a maternal and newborn health and survival related problem in their respective clusters and re-plan for continued improvement

of their own response to the problem such as coordinating economic and family support when a mother has to travel to health care [22]. In addition, the intervention will benefit from the process-based approaches of Theory of Change (TOC) principles [23]. TOC model helps to understand and implement how activities, outcomes, and impacts are connected to each other. In our intervention, this means processes such as identifying mothers and newborns at risk of falling from essential and emergency services, educating and preparing at household level, creating a supportive mechanism when referral is needed, and emergency support at home. TOC connects these activities with outcomes such as improved continuum of care completion and impacts such as reduced maternal and newborn mortalities. Furthermore, WHO and Ethiopian Ministry of health principles of essential maternal and newborn care frameworks will be implemented [24,25] to improve the continuum of care.

### Trial schedule: SPIRIT schedule (Fig 2 and Table 2)

### Ethical approval

The study received a renewal of approval from the Institutional Review Board (IRB) of Hawassa University in Ethiopia (IRB/363/15) on August 5, 2024, extending the approval until

| | Enrolment | Intervention | Follow up | | | | |
|---|---|---|---|---|---|---|---|
| **TIMEPOINT** | *October 1/2023/* | Sept 1 2023/Nov 202024 | *October 1/2023* | *t₂* | *t₃* | *t₄* | *FEB 28,2025* |
| **ENROLMENT:** | | | | | | | |
| Screening and enrollment of study participants | X | | | | | | |
| Implementation of the Intervention | | | ▬ | | | | |
| Follow up | | | | ▬ | | | |
| Assessment and Collecting data **SESSMENTS:** | | | | | | | |
| *[Pre intervention of continuum of care]* | X | | | | | | |
| *[continuum of maternal and neonatal care]* | | | X | X | X | X | X |

**Fig 2. Schedule of enrolment, interventions, and assessments cluster randomized controlled trial to assess the effectiveness of a package of community-based intervention on continuum of maternal and newborn healthcare in Sidama, Ethiopia:The SiMaNeH trial protocol.**

Table 2. Timeline of the trial (SPIRIT guideline compliant).

| Timeline | Enrolment | Allocation | Pre-recruitment | Recruitment | Follow-up |
|---|---|---|---|---|---|
| Oct-Dec 2023 | X | | | | |
| Jan 2024 | | X | | | |
| Feb-Jun 2024 | | | X | | |
| Jun-Sept 2024 | | | | X | |
| Oct 2024-Feb 2025 | | | | | X |

August 6, 2025. A support letter was obtained from the Sidama Regional State Health Bureau and relevant local authorities.

All participants will be provided with a written consent information sheet, which will be attached to this protocol. They will be informed about the aims of the study, as well as their right to refuse participation or withdraw from the trial at any time without any consequences

## Study design

This study will use a cluster randomized controlled trial with two arms of equal size. We use villages (kebeles in Ethiopian structure) with an average population of about 5,000 people as the clusters served as units of randomization.

## Randomization technique and blinding

We selected 20 clusters randomly from eligible 80 from four districts in the region and allocated them into intervention and control in 1:1 allocation,10 each to the intervention and control cluster. To balance based on the distance of clusters from nearby health facilities between intervention and control clusters, we stratified first the 20 clusters into four categories based on the population size and distance to the nearest health facility to reduce confounding. The four strata were: 1) Higher population (over 4,000 residents) and longer distance (more than an hour of walking distance), 2) Higher population and shorter distance (less than an hour walking distance 3) Lower population (less 4,000 residents) and shorter distance and 4) Lower population and longer distance. Then clusters from each stratum were randomly allocated to intervention and control using a computer-generated random number using simple randomization. A seed number was produced and sent from the University of Bergen as a starting point for random number generation. Blinding is difficult because of the nature of cluster randomized trials, but we created a buffer zone between intervention and control clusters to reduce intervention contamination. Data collectors are independent of the intervention process and do not have formal information on which study arm the cluster to which they collect data belongs.

## Sampling strategy and sample size

To calculate the sample size of the participating mothers and babies in the study, we used the sample size calculation for cluster randomized controlled trials with the fixed number of clusters recommended by Karla Hemming and colleagues [26]. We fixed the number of clusters to 20 for logistic and access reasons. We base our estimate of pregnant women with annual crude birth rate of 30 births per 1000 population per United Nations Population Division projection for Ethiopia in 2024 [20]. The average pregnant mother-baby pair per cluster was estimated to be 150 during the study period. To estimate the minimum required sample, the trial aims to increase the completion rate of the continuum of pregnancy, birth, and postpartum care from an estimated 20% exiting status to 30% in intervention clusters with a 95% significance level and 80% power. We take the intra-cluster correlation of 0.01 [1]. Substituting these numbers

in the following formula gives the minimum number of mother-baby pairs for the study. We used the following formula to calculate the sample size.

$$n = \frac{\left(Z\alpha/2 + Z\beta\right)^2 \times \left(p1\left(1-p1\right) + p2\left(1-p2\right)\right)}{\left(p1-p2\right)^2} \times \left(1+m-1\right) \times ICC$$

- $n$ is the sample size = number of pregnant mother-baby pairs needed per study arm

- $Z_{\alpha/2}$ is the z-value corresponding to the significance level ($\alpha$=0.05, $Z\alpha/2$ =1.96).

- $Z_{\beta}$ is the z-value corresponding to the desired power ($\beta$ = 0.20, $Z\beta$ = 0.84).

- $p_1$ is the expected proportion in the control group (0.20), the continuum of care completion in control clusters.

- $p_2$ is the expected proportion in the intervention group (0.30), expected continuum of care completion in intervention clusters.

- $m$ is the average cluster size = 150 mothers per cluster on average.

- $ICC$ is the intra-cluster correlation coefficient (0.01).

This provides a minimum of 723 pregnant women-baby pairs per trial arm, resulting in a total sample size of 1446. However, because several outcome measures may need more samples and to have adequate numbers for sub-analysis, we decided to recruit a total of 2000 woman-baby pairs (with addition of 38% participants), 1000 in each arm.

## Participants

**Inclusion criteria.** We shall include mothers who permanently reside in the study clusters and consent to participate. All pregnant mothers who fulfill the above criteria and do not meet the following exclusion criteria will participate. This includes mothers in their gestational period of between the 20th and 26th weeks within the four months of the recruitment period (June-September 2024).

**Exclusion criteria.** We will not recruit mothers with less 20th weeks of gestation because of longer time needed for follow-up. In addition, mothers whose pregnancy terminated before 20th week of gestation will not be included because the main emphasis of the study is on completion of maternal care and utilization of newborn care. Mothers who do not consent to participate in the study after information will not be included.

## Consultative meeting

Pre-intervention there a one day consultative meeting with Sidama Regional Health Bureaus, maternal health department, health extension department staff, and the same structure representatives from district health office staffs at Hawassa. The aim of this meeting is to introduce the study and create a fertile ground in a communication with the administrative bodies. The other issue is to get general agreement on the staff turnover, health extension workers who received the intervention and engaged in the research process will not be transferred until the period of study duration. The administrative bodies' involvement makes the community and health care providers positive for the implementation of the intervention.

## Launching ceremony

Launching ceremony was held following the consultative meeting in the same week. In this ceremony community representative from each intervention and control kebeles two

representatives (head of the kebele and one person from kebele council or respected community leader) was introduced about the study. The aim of this ceremony is for community involvement that is crucial for the implementation and the research process. This was considered as a community based consent for intervention and the whole data collection both in the intervention and control group. The other importance for intervention clusters, if there is problem in the implementation by health extension worker the kebele leader and the council were given responsibility to motivate for better performance.

## Intervention

To achieve the aim of improving the completion rate of the continuum of essential care from pregnancy care through postnatal care, the intervention package uses four mechanisms of action to help reduce barriers to essential healthcare utilization. The four mechanisms of action are: 1) Actively identify and connect to healthcare services, 2) Educate and prepare at mothers and family members home, 2) Support emotionally and materially through local collective efforts (COLs), and 4) Provide home-based essential care in case care from health facilities is not actual.

To implement the four mechanisms of action stated above, the project coordinates the efforts of three existing community-based stakeholder groups through training and supervision. The three stakeholders are the community health workers (HEWs in Ethiopia), the women support groups (HDTs in Ethiopia), and the community resources groups, also termed community opinion leaders (COLs).

HEWs are women permanently employed on average two in all clusters (Kebeles) in Ethiopia and paid monthly salary from the government. They have overall responsibility in their clusters for health promotion, disease prevention, and provision of basic maternal and newborn care such as antenatal, postnatal, and baby resuscitation. HEWs will participate majorly in three of the four mechanisms of action, namely identification, educating and preparing, and provision of home-based essential care when delivery happens at home, in addition to routine postnatal care. Furthermore, they will coordinate the work of COLs for the local collective support efforts.

HDTs are about 30 voluntary groups of women in each cluster residing in sub-villages of clusters in Ethiopia, freely helping the works of HEWs. They have recognition from the government without getting paid. Because of their unique position closer to sub-villages, they have the unique opportunity to identify mothers as they become pregnant in their respective villages and connect to healthcare services. They also educate and prepare mothers and families for standard and emergency care. HDTs' main contribution to the intervention is actively identifying and connecting mothers and babies to essential standard care and emergency care when they observe danger signs. HDTs will give particular attention to high-risk and hard-to-reach mothers and newborn babies such as extremely poor, living in difficult topography and distant villages from the health facilities and HEWs stations as well as those with existing sicknesses and malnutrition. As such, the contribution of the village HDTs in the four mechanisms of action will be identifying and connecting, education and preparing and involving with COLs in strengthening financial, transport, and family care support through collective local efforts so that a mother or baby reaches to health facility and receives care.

Community opinion leaders (COLs) are key leaders who lead opinions such as approval of the importance of receiving healthcare services and coordinating local collective help mechanisms. COLs are a group of local leaders forming a committee which includes influential religious leaders, cultural leaders, local government administrative leaders, and leaders of local welfare scheme called Idir in Ethiopia. Idir is a local welfare system where families contribute

financially or materially every month. This welfare system often shares serious costs during deaths and severe sicknesses.

The main responsibility of the COLs is identifying problems such as barriers hindering mothers and babies receiving important health care and falling out of the continuum of care. They respond to the problem by coordinating local collective efforts, for example transport or financial support to reach a health facility when a mother or baby is referred to and taking care of the family members at home. For this contribution, they meet every two-weeks to evaluate and re-plan their actions based on the principles of Participatory Learning and Action (PLA) model [21]. HDTs help COLs with gathering and providing sub-village information. COLs use their high level of acceptance to visit and support families where the relationship between the woman and her husband is not smooth and put great pressure on the health and wellbeing of women or their newborns. At such households, COLs provide essential advice on the importance of care for a pregnant women and newborn babies. This is expected to contribute to a mother and baby receiving essential care that improve their survival and wellbeing.

The project provides training and supervision support to the three key stakeholders participating in intervention.

For COLs, the training and supervision will be based on PLA model of problem identification, providing local solutions, evaluating the effect, and re-planning process. COLs receive day-long initial training and a two-monthly supervision discussion for an hour during the project period.

Training and supervision to HEWs will have two dimensions: On one hand, HEWs will be trained and supported on technical refresher issues related to baby resuscitation and home-based immediate support when a birth occurs at home, and essential postnatal maternal and newborn care and on the other hand, they will be trained in leadership role to lead the efforts of COLs and HDTs. HEWs receive two days of intensive training followed by two-monthly supervision and monitoring meetings with the project team.

The training and supervision of HDTs will be mainly on identifying and connecting mothers and babies for normal essential care as well as detecting danger signs and referring to emergency care babies and mothers with severe sickness (danger signs). In addition, HDTs and COLs receive training on maternal essential self-care, preparation for healthcare, nutrition, rest, hygiene, and other important information which in turn they will teach mothers and families. Further, the HDTs receive training in tracking and giving special attention to high-risk mothers and babies in difficult families and hard-to-reach villages so that they are not left out or fall out of the critically needed healthcare services.

## Training materials used in the intervention training

To ensure the intervention is delivered the same way across individuals (mothers and support groups) and clusters, we provided printed materials and an 8-minute voice recording, both in the local Sidama language. A female speaker with the language as the mother tongue recorded the message, and both the print and audio content cover the same topics.

1) Content for Maternal and Newborn Care:

- **Antenatal Care**: Importance of starting care before three months of pregnancy and use of the recommended contact without discontinuation

- **Pregnancy Danger Signs**: What to watch during pregnancy

- **Birth Preparedness complication readiness plan**: Making plans for complications and delivery

- **Institutional delivery**: Encouraging all women to give birth at a health facility

- **Maternity waiting room**: Availability and benefits

- **Postnatal care**: Importance, timing, and warning signs after childbirth

2) **Content for newborn Care:**

- **Thermal care**: Keeping the newborn warm at home

- **Breastfeeding**: Guidelines for proper feeding

- **Neonatal danger signs**: how to identify problems early and referring

The contents of the training materials are used during training sessions and revised during every supervision meeting with project staff. In the community, education is given to pregnant women at their homes, in churches, or during women's forums.

**Home visits and support.** Every two weeks, a Community Opinion Leader (COL) member and Health Development Team woman (HDT) visit pregnant women at their homes. The pairing COL-HDT considers at least one of them having literacy and owning a mobile phone with enough storage for the voice recordings used as teaching guides. After home-visits, these support groups bring to their common two-weekly PLA forum to evaluate and re-plan.

Through these mechanisms, the project builds a supportive network around mothers to help improve the initiation and completion of essential services in the continuum of care. Table 3 below describes the roles of each stakeholder to the four mechanisms of actions to improve continuum of care in the intervention clusters.

The ideal continuum of care can extend to pre-conception care that includes care for young girls before they even become pregnant and continue forward until the newborn becomes a

**Table 3. The role of three different stakeholders for the four community-based intervention actions (strategies) in intervention cluster, Sidama regional State, Ethiopia.**

|  | 1) Identify and connect | 2) Educate and prepare | 3) Support through collective efforts | 4) Home-based essential care |
|---|---|---|---|---|
| HEWs | All pregnant mothers for essential ANC, delivery, and postnatal care in collaboration with HDTs and COLs Mothers and babies in high-risk of falling out of care, or even do not initiate first contact. Screen and recruit mothers for study 20–26th week of gestation. | Mothers and families on essential care, nutrition, hygiene… Prepare mothers and family for facility delivery, emergency health seeking. | Mainly through coordinating the efforts of COLs | Encourage antenatal care as per the guidelines of MoH Emergency maternal and newborn help for deliveries occurred at home. Routine postnatal care for mothers and babies. |
| HDTs | Identify mothers and babies in her sub-village and connect to HEWs, COLs and health care. Guide data collectors to locate homes. | Educate and prepare mothers and families on essential care, nutrition, hygiene, emergency care | Inform to COL s and HEWs about mothers and babies not receiving care | Educate and prepare. Do not provide any technical care. |
| COLs | Identify mothers and babies at high-risk and hard-to-reach villages. Identify mothers and babies in families with non-smooth relationships and in dangers. Connect those identified to healthcare services. | Approve and promote the importance of essential health care. Make PLA discussion to prepare for support when needed. | Facilitate finance, transport, and family care coordinating local collective efforts. Continuously monitor their support provision | Make home visits to difficult families and provide advice and emotional support. |

*Note: HEWs = Health extension workers, HDTs = Health development team, COLs = Community opinion leaders (religious, cultural, local welfare scheme, and administrative leaders working as a group to ensure support mechanisms), PLA = Participatory Learning and Action (a community problem solving model)*

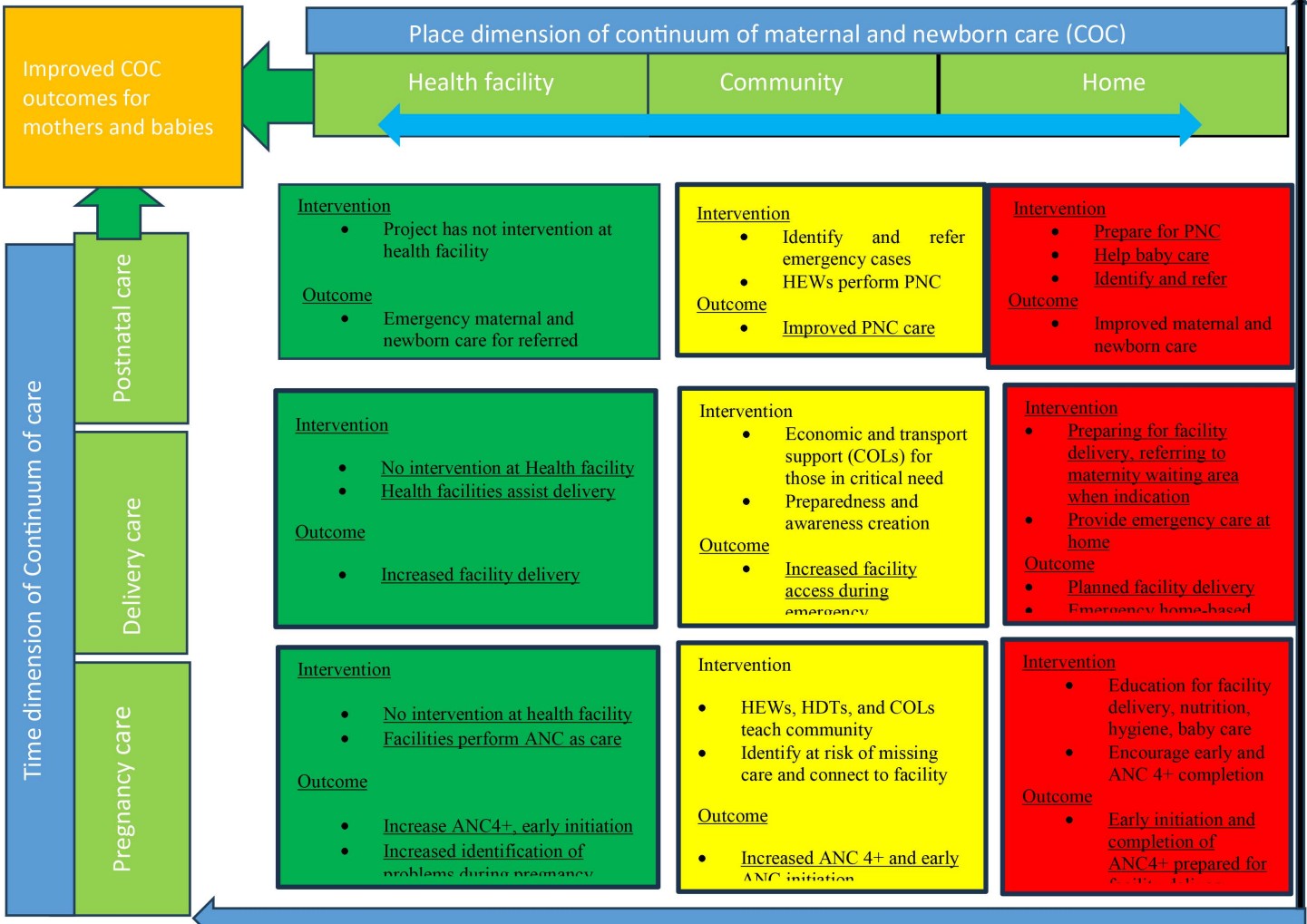

**Fig 3. Conceptual framework of the intervention in relation to the time and place dimensions of continuum of maternal and newborn care (COC).** Authors created this figure and give permission under creative commons CC BY 4.0 license to all who want to use the image. **Note**: The Conceptual framework is prepared based on several published sources that described the concept and dimensions of continuum of maternal and newborn care [18,25,27]. Fig 3.

grown child. However, for logistic and practical purposes we limit the time dimension from pregnancy to postnatal period in this trial.

## Outcome and process indicators

Outcome measures will not be targeting the effect of each component of the intervention strategies separately. Rather, we will measure the overall collective and cumulative effect of these intervention strategies in improving continuum of maternal care and essential newborn care outcomes.

The primary maternal outcome measure is the completion rate of essential maternal care. We define a continuum of care completion as follows:

1) fully completed care when a mother has received four or more antenatal care, delivered in a health facility with a skilled attendant, and received at least four postnatal care (first 24 hours, 48–72 hours, 7–14 days, and six weeks after delivery) from HEWs or other health

professionals. We will also measure and report the outcomes (effects of intervention) separately for antenatal care 4+ completion, delivery at skilled health facility and at least four contacts of postnatal care.

2) Emergency home-based care when the delivery occurred at home and mother and baby received immediate visit at home by HEWs or a higher health professional.

3) Sub-analysis on the primary outcome will compare intervention with control clusters for assistance received from HEWs or other professionals at home when a birth occurred at home, and proportion of mothers who stayed at maternal-waiting shelters immediate days before delivery.

4) Proportion received financial, transport, and family care support from community opinion leaders (COLs) when support is critically needed.

5) The proportion of newborn babies received the four essential newborn care services (immediate and thorough drying, skin-to-skin contact, immediate breastfeeding initiation, and delayed cord clamping). Because completion of maternal continuum of care (COC) has an implication for newborn essential care utilization and survival, we will use maternal COC as an exposure variable for essential newborn care utilization. In other words, we will describe whether maternal completion or not completion of COC has a significant effect on utilization of essential newborn care indicators.

6) Emergency newborn sickness identification and referral rates.

7) Secondary outcome measure will be on whether the intervention influenced neonatal mortality and stillbirths.

For newborn essential care outcomes, we will measure and compare with the control arms the proportion of babies immediately put to breastfeeding and exclusively feeding until at least a month after birth, the proportion of babies received essential thermal care (wrapping in close, skin-to-skin contact to mother immediately, delayed baby bathing), baby resuscitation at home and in clinics, and proportion of babies received emergency treatment and referrals when they had danger signs.

## Data collection

Data will be collected by 20 trained independent data collectors. Data collectors are people with educational backgrounds of a high-school complete or more and selected from the districts where data were collected so that they understand the cultural tricks and language of the people better. Data collectors will receive a week-long training, including piloting the questionnaire in selected other clusters closer to the study area. There will be a supervisor per five data collectors (a total of four) to ensure the completeness and reliability of data. Data will be collected using a structured questionnaire developed including well recognized variables that capture maternal and newborn continuum of care information through face-to-face interview with mothers or other care givers and health facility review. We will use koboCollect, a tablet-based application which also incorporates Geographical positioning system (GPS) to collect data.

All four strategies of our intervention are intended to contribute to pregnant mothers navigating through all essential and emergency continuum of care which also has an implication to essential newborn care utilization and emergency care. As such we include questions in the data collection that target whether a mother and newborn baby received adequate recommended pregnancy, childbirth, and postnatal care and newborn care which are the

outcomes of the study. As such we do not target questions to measure specific contribution of a particular intervention. We measure the cumulative effect of the package of intervention. However, we included questions specifying whether the mother or the newborn has received any economic or family care support if there was referral. This will be presented in the result as descriptive data.

We will collect data in five rounds at households and health facilities after the intervention is implemented. First, we will collect data when the mother's gestation is 20th to 26th week when she formally recruited for subsequent follow-up. This period matches with the time when a mother became pregnant after the start of the intervention process in the intervention cluster. In other words, the mothers who enter the first data collection and follow-up phase on their 20th to 26th week of gestation have been exposed to interventions since they were pregnant. In this first round of data collection, data will be collected on basic socio-demographic and important variables for important covariates such as wealth status indicators, distance to health facility, history of previous pregnancy and delivery, healthcare utilization during the previous pregnancy and delivery. Additionally, whether the mother has initiated antenatal care in this pregnancy and when, where and from who she is getting the service, any danger sign experienced in the current pregnancy and her own knowledge of early initiation and importance of antenatal care, knowledge on danger signs and preparedness related questions. In this phase data will be collected at household level with the mother as primary respondent. However, when the mother does not remember or cannot answer, the record will be checked at health facility if she had a visit. For triangulation ANC initiation time and number of visit she received will be collected from health facility record.

Follow-up data collection will include the second to the fifth rounds of data collection. The second round will be 10 weeks after the first data collection (30th to 36th week of gestation). This data collection will mainly focus on information about variables related to the whether the mother continued to receive antenatal care services, whether she had pregnancy-related complications and severe sickness, and information on healthcare received or not for those who had sickness. In addition, whether the community support group had helped the mother. The place of data collection and source of data will be like first round of data collection above.

The third round of data collection is on the fourth day after delivery. The main variables in this round of data collection include variables on antenatal care completion, information on delivery care such as place of delivery, delivery attendant, utilization of postnatal care during the WHO recommended first 24 hours and 48–72 hours after birth, whether the mother received any financial, transport, or family care support from the COLs system to reach to health facility, whether the mother stayed in the maternal waiting shelter nearby health facilities the immediate days before delivery, delivery outcome baby and maternal survival and morbidity. In this round of data collection information will include both maternal and newborn essential care and complications as well as measures taken, essential newborn care provided during delivery (resuscitation, umbilical care, thermal care, initiation of breastfeeding, treatment for severely sick and premature babies). Most of data collection in this phase will be from the records of health institutions in addition to interviews to the mother, if the delivery takes place at a health facility.

The fourth round of data will be collected two weeks after delivery (10 days after the third data collection). Data in this round will include the first and second week maternal and newborn survival, occurrence of morbidity, and emergency help received for those who had serious sickness. Additional information will be collected whether the mother and baby had received postnatal care during the WHO recommended 7–14 days after birth, and essential baby care within the first two weeks. The sources of data for this fourth round of data collection will be mothers or any adult family member. However, when there is a doubt on the

validity of the data, health facility records will be cross-checked. The final and fifth round of data collection will take place six weeks after delivery. This data collection will include variables on the first month and six weeks essential postnatal maternal and neonatal care, first month and six-week survival of the mother and newborn, and information on sickness and emergency healthcare received for sickness in this period.

## Data storage and security plan

Data will be transferred from koboCollect, a tablet-based application, directly to the data server at Hawassa University research center. Only people involved in the project will have password-based access. Data will be de-identified of personal identifiers when stored. Personal variables will never be published anywhere. However, deidentified data will be available to researchers when requested with clear plan for use as stated in the protocol registration. Data will be stored for five years after the completion of the study and will be deleted in December 2030.

## Data entry and analysis plan

Data will be double entered into excel sheet and transferred to R software version 4.4.1for analysis [28]. Data will be analyzed and presented in a format comparing outcomes between intervention and control arms using intention-to-treat analysis, meaning all recruited will be analyzed irrespective of finishing the study process. Data analysis will use techniques that consider the clustered nature of the data. That means we will use multilevel regression analysis through Generalized Linear Mixed Models (GLMMs) with binary logistic regression to include random effects of the clustering at district level on the outcomes. We will particularly determine and report intra-cluster correlation that quantifies the random effect. For fixed effects, the main variable is the cumulative effect of the intervention (intervention itself). However, we will also include fixed effects covariates such as individual level maternal variables such as parity, age, education wealth index and environmental fixed effects variables such as distance to the health facility, cluster level of education of mothers, cluster's level of wealth index (for example, the percentage of mothers in the poorest wealth index in the cluster), support from COLs during emergencies, and other fixed effects individual and environmental variables.

## Cost-effectiveness analysis

We will consider only intervention-related costs such as transport, training, data collectors and supervisors' payments, costs related to project staff. The effectiveness data will be the relative change in essential newborn care utilization in intervention clusters compared to control clusters. We will basically analyze and report Incremental Cost-effectiveness Ratio (ICER).

## Dissemination plan

We will communicate the results with local government structures, Sidama Regional State Health Bureau and publish scientific articles in internationally recognized peer-reviewed journals.

## Strengths and limitations

The primary strength of this trial lies in its innovative approach that implement comprehensive strategies involving a wide range of key community stakeholders. Ethiopian government system has a strong local structure with important stakeholder to support health interventions. These stakeholders include religious, cultural, administrative, and local welfare scheme leaders, who lead and influence opinions and collective resources in rural communities. They

will work collaboratively alongside the community health workers and women's support groups. To improve initiation and continuum in the care, this trial proposes an integrative approach aimed at actively identifying and connecting women and newborns to routine and emergency healthcare services, with increased emphasis on women and newborns in vulnerable households and hard-to-reach villages.

The strategy focuses on strengthening existing local mechanisms to ensure initiation and continuity of essential maternal and newborn care. This includes facilitating access to finance, transport, and family care, in addition to emotional support thereby addressing critical barriers to healthcare access.

Despite the strengths described above, the trial may face limitations due to its focus primarily on the quantitative improvement of healthcare utilization, without addressing challenges related to the quality of healthcare. The quality of healthcare measured through quality of provision and the personal experiences of service recipients are crucial factors influencing the continued utilization of healthcare services. However, given the limited scope and resources, the current trial prioritizes the initial step of connecting individuals to healthcare which in turn has the potential to create pressure demanding improved quality of care. We believe that creating demand through such as our trials may lead further to interventions specifically targeting the improvement of healthcare quality.

## Supporting information

**S1 File. Study Protocol (Study_Protocol.docx) – Detailed description of the study design, methodology, and procedures.**
(DOCX)

## Author contributions

**Conceptualization:** Achamyelesh Gebretsadik, Yemisrach Shiferaw.

**Formal analysis:** Achamyelesh Gebretsadik.

**Methodology:** Achamyelesh Gebretsadik, Yemisrach Shiferaw, Hirut Gemeda, Yaliso Yaya.

**Project administration:** Achamyelesh Gebretsadik.

**Software:** Hirut Gemeda.

**Supervision:** Achamyelesh Gebretsadik, Yemisrach Shiferaw, Hirut Gemeda, Yaliso Yaya.

**Validation:** Yemisrach Shiferaw, Hirut Gemeda.

**Visualization:** Yaliso Yaya.

**Writing – original draft:** Achamyelesh Gebretsadik, Yaliso Yaya.

**Writing – review & editing:** Yemisrach Shiferaw, Hirut Gemeda, Yaliso Yaya.

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
