## [Decision Letter · Decision Letter 0]

7 Oct 2024

PONE-D-24-34008Cluster randomized controlled trial to assess the effectiveness of a package of community-based intervention on continuum of maternal and newborn healthcare in Sidama, Ethiopia:The SiMaNeH trial protocol.PLOS ONE

Dear Dr. Tekle,

Thank you for submitting your manuscript to PLOS ONE. After careful consideration, we feel that it has merit but does not fully meet PLOS ONE’s publication criteria as it currently stands. Therefore, we invite you to submit a revised version of the manuscript that addresses the points raised during the review process.

We look forward to receiving your revised manuscript.

Kind regards,

Anteneh Fikrie, MPH

Academic Editor

PLOS ONE

Journal Requirements:

1. When submitting your revision, we need you to address these additional requirements. Please ensure that your manuscript meets PLOS ONE's style requirements, including those for file naming. The PLOS ONE style templates can be found at https://journals.plos.org/plosone/s/file?id=wjVg/PLOSOne_formatting_sample_main_body.pdf and https://journals.plos.org/plosone/s/file?id=ba62/PLOSOne_formatting_sample_title_authors_affiliations.pdf 2. We suggest you thoroughly copyedit your manuscript for language usage, spelling, and grammar. If you do not know anyone who can help you do this, you may wish to consider employing a professional scientific editing service.  The American Journal Experts (AJE) (https://www.aje.com/) is one such service that has extensive experience helping authors meet PLOS guidelines and can provide language editing, translation, manuscript formatting, and figure formatting to ensure your manuscript meets our submission guidelines. Please note that having the manuscript copyedited by AJE or any other editing services does not guarantee selection for peer review or acceptance for publication.  Upon resubmission, please provide the following: The name of the colleague or the details of the professional service that edited your manuscript A copy of your manuscript showing your changes by either highlighting them or using track changes (uploaded as a *supporting information* file) A clean copy of the edited manuscript (uploaded as the new *manuscript* file) 3. When completing the data availability statement of the submission form, you indicated that you will make your data available on acceptance. We strongly recommend all authors decide on a data sharing plan before acceptance, as the process can be lengthy and hold up publication timelines. Please note that, though access restrictions are acceptable now, your entire data will need to be made freely accessible if your manuscript is accepted for publication. This policy applies to all data except where public deposition would breach compliance with the protocol approved by your research ethics board. If you are unable to adhere to our open data policy, please kindly revise your statement to explain your reasoning and we will seek the editor's input on an exemption. Please be assured that, once you have provided your new statement, the assessment of your exemption will not hold up the peer review process. 4. We note that Figure 2 in your submission contain [map/satellite] images which may be copyrighted. All PLOS content is published under the Creative Commons Attribution License (CC BY 4.0), which means that the manuscript, images, and Supporting Information files will be freely available online, and any third party is permitted to access, download, copy, distribute, and use these materials in any way, even commercially, with proper attribution. For these reasons, we cannot publish previously copyrighted maps or satellite images created using proprietary data, such as Google software (Google Maps, Street View, and Earth). For more information, see our copyright guidelines: http://journals.plos.org/plosone/s/licenses-and-copyright. We require you to either (1) present written permission from the copyright holder to publish these figures specifically under the CC BY 4.0 license, or (2) remove the figures from your submission: 1. You may seek permission from the original copyright holder of Figure 2 to publish the content specifically under the CC BY 4.0 license.   We recommend that you contact the original copyright holder with the Content Permission Form (http://journals.plos.org/plosone/s/file?id=7c09/content-permission-form.pdf) and the following text:“I request permission for the open-access journal PLOS ONE to publish XXX under the Creative Commons Attribution License (CCAL) CC BY 4.0 (http://creativecommons.org/licenses/by/4.0/). Please be aware that this license allows unrestricted use and distribution, even commercially, by third parties. Please reply and provide explicit written permission to publish XXX under a CC BY license and complete the attached form.” Please upload the completed Content Permission Form or other proof of granted permissions as an ""Other"" file with your submission. In the figure caption of the copyrighted figure, please include the following text: “Reprinted from [ref] under a CC BY license, with permission from [name of publisher], original copyright [original copyright year].” 2. If you are unable to obtain permission from the original copyright holder to publish these figures under the CC BY 4.0 license or if the copyright holder’s requirements are incompatible with the CC BY 4.0 license, please either i) remove the figure or ii) supply a replacement figure that complies with the CC BY 4.0 license. Please check copyright information on all replacement figures and update the figure caption with source information. If applicable, please specify in the figure caption text when a figure is similar but not identical to the original image and is therefore for illustrative purposes only.The following resources for replacing copyrighted map figures may be helpful: USGS National Map Viewer (public domain): http://viewer.nationalmap.gov/viewer/The Gateway to Astronaut Photography of Earth (public domain): http://eol.jsc.nasa.gov/sseop/clickmap/Maps at the CIA (public domain): https://www.cia.gov/library/publications/the-world-factbook/index.html and https://www.cia.gov/library/publications/cia-maps-publications/index.htmlNASA Earth Observatory (public domain): http://earthobservatory.nasa.gov/Landsat:
http://landsat.visibleearth.nasa.gov/USGS EROS (Earth Resources Observatory and Science (EROS) Center) (public domain): http://eros.usgs.gov/#Natural Earth (public domain): http://www.naturalearthdata.com/

Reviewers' comments:

Reviewer's Responses to Questions

**Comments to the Author**

1. Does the manuscript provide a valid rationale for the proposed study, with clearly identified and justified research questions?

Reviewer #1: Yes

Reviewer #2: Yes

2. Is the protocol technically sound and planned in a manner that will lead to a meaningful outcome and allow testing the stated hypotheses?

Reviewer #1: Partly

Reviewer #2: Yes

3. Is the methodology feasible and described in sufficient detail to allow the work to be replicable?

Reviewer #1: No

Reviewer #2: Yes

4. Have the authors described where all data underlying the findings will be made available when the study is complete?

Reviewer #1: Yes

Reviewer #2: Yes

5. Is the manuscript presented in an intelligible fashion and written in standard English?

Reviewer #1: Yes

Reviewer #2: Yes

6. Review Comments to the Author

You may also provide optional suggestions and comments to authors that they might find helpful in planning their study.

Reviewer #1: Thank you for your submission. The justification for your protocol is clear the aims also clearly identified. For a more robust proposal, I would suggest the following:

1. Write the data collection details with under the headings of each intervention. This will include details on how (and where) the data will be collected, which tool, and what information will be sought. Same for the data analysis section.

2. Include an international framework that can be cited for the choice the interventions.

3. Report the increase in sample size from 1446 to 2000 as a percentage.

4. Plan to establish a memorandum of understanding with community partners. It is great that there is such a good community base of networks – it is important to have a memorandum of understanding.

5. Indicate how long the data will be stored

6. Consider using multilevel models to account for individual levels

7. Describe which cluster level analyses will be conducted

8. Figure 1, part seems to have lost formatting

9. Indicate how you will evaluate cost effectiveness. Is there a health economist on the team?

Reviewer #2: I congratulate the authors for their dedication and meaningful contribution to the study. In general, I suggest submitting the protocol to Editage or another reputable language editing service for further refinement.

7. PLOS authors have the option to publish the peer review history of their article (what does this mean? ). If published, this will include your full peer review and any attached files.

**Do you want your identity to be public for this peer review?** For information about this choice, including consent withdrawal, please see our Privacy Policy .

Reviewer #1: No

Reviewer #2: **Yes: ** Alo Edin

---

## [Author Response · Author response to Decision Letter 1]

1 Nov 2024

Date 1/11/2024

Point to point Responses for reviewer comments

We really appreciate your constructive and insightful comments.

• Editor

Comments to Author Response

Please amend the title either on the online submission form or in your so that they are identical. Thank you for your comments. It is done

When completing the data availability statement of the submission form, you indicated that you will make your data available on acceptance. We strongly recommend all authors decide on a data sharing plan before acceptance, as the process can be lengthy and hold up publication timelines. Please note that, though access restrictions are acceptable now, your entire data will need to be made freely accessible if your manuscript is accepted for publication. This policy applies to all data except where public deposition would breach compliance with the protocol approved by your research ethics board. If you are unable to adhere to our open data policy, please kindly revise your statement to explain your reasoning and we will seek the editor's input on an exemption. Please be assured that, once you have provided your new statement, the assessment of your exemption will not hold up the peer review process. Dear Editor,

Thank you very much for raising this important point. As this is a proposal, we do not have data at hand. However, once we collect the data and submit a manuscript for publication, we will be happy to provide it upon request.

Please ensure that you refer to Table 1 in your text as, if accepted, production will need this reference to link the reader to the Table. Thank you table 1 already cited in the text.

The concern of the journal office is not Figure 1 as a whole, but rather the underlying map image used to construct Figure 1. Please specify the source of the map used to construct Figure 1 and its copyright status.

Thank you for your request. The authors created the map using ArcGIS software. The shape file is freely available online; however, it originates from the Central Statistical Agency, and we have cited that source in the map.

---

## [Decision Letter · Decision Letter 1]

15 Dec 2024

PONE-D-24-34008R1Cluster randomized controlled trial to assess the effectiveness of a package of community-based intervention on continuum of maternal and newborn health care in Sidama, Ethiopia:The SiMaNeH trial protocol.PLOS ONE

Dear Dr. Tekle,

Thank you for submitting your manuscript to PLOS ONE. After careful consideration, we feel that it has merit but does not fully meet PLOS ONE’s publication criteria as it currently stands. Therefore, we invite you to submit a revised version of the manuscript that addresses the points raised during the review process.

We look forward to receiving your revised manuscript.

Kind regards,

Anteneh Fikrie, MPH

Academic Editor

PLOS ONE

Journal Requirements:

Reviewers' comments:

Reviewer's Responses to Questions

**Comments to the Author**

1. Does the manuscript provide a valid rationale for the proposed study, with clearly identified and justified research questions?

Reviewer #3: Yes

2. Is the protocol technically sound and planned in a manner that will lead to a meaningful outcome and allow testing the stated hypotheses?

Reviewer #3: Partly

3. Is the methodology feasible and described in sufficient detail to allow the work to be replicable?

Reviewer #3: Yes

4. Have the authors described where all data underlying the findings will be made available when the study is complete?

Reviewer #3: No

5. Is the manuscript presented in an intelligible fashion and written in standard English?

Reviewer #3: No

6. Review Comments to the Author

You may also provide optional suggestions and comments to authors that they might find helpful in planning their study.

Reviewer #3: I come to this revised protocol as a new reviewer. The protocol is for a cluster randomised trial of community-based interventions in Ethiopia.

There are couple of infelicities in the English - eg the spare word Ethiopia in the abstract (end of line 4 or the background; tense on line 3-4 of setting part of methods etc). Please carefully copyedit the manuscript.

I'm unclear in Table 1 how this relates to Figure 1 - is this the expected birthrate of the red zone or of the purple and yellow zones. This needs moer explanation, and the sizes of the different Kebeles given to ensure proper balance and range of sizes. The buffer zones - and the fact there are not universally buffers between Kebeles in different arms needs to be explained as well as Figure 1 is introduced.

Please justify an ICC of 0.01 - where has this been taken from, and please demionstrate it is relevant to this setting.

We are told that 2000 pairs will be used here, but 150 per cluster - please discuss the discrepancy here.

There is an issue with individual consent and cluster randomisation in terms of differential consent by arm - one generally needs to explain how this is to be guarded against and what the expected consent rates are going to be - will willingness to consent increase the ICC? Is there any way of getting some data from routine records? Completion of visits is presumably routinely reported enabling an ITT analysis.

The issue of precisely what data is to be collected and how analysed is missing.

7. PLOS authors have the option to publish the peer review history of their article (what does this mean? ). If published, this will include your full peer review and any attached files.

**Do you want your identity to be public for this peer review?** For information about this choice, including consent withdrawal, please see our Privacy Policy .

Reviewer #3: No

---

## [Author Response · Author response to Decision Letter 2]

21 Jan 2025

Dear Editor,

Thank you for providing us additional comment by inviting third new reviewer and for resubmission of our manuscript PONE-D-24-34008 entitled “Cluster randomized controlled trial to assess the effectiveness of a package of community-based intervention on continuum of maternal and newborn healthcare in Sidama, Ethiopia: The SiMaNeH trial protocol.”

We appreciate the new observation of on our manuscript which makes the protocol more clear and understandable to the scientific community.

Below follows a point-by-point rebuttal to each comment from the reviewer.

Reviewer comment

There are couple of infelicities in the English - eg the spare word Ethiopia in the abstract (end of line 4 or the background; tense on line 3-4 of setting part of methods etc). Please carefully copyedit the manuscript

Response: Thank you for your comment as per your suggestion English is revised in the abstract and background section incudiding the entire document

Reviewer comment

I'm unclear in Table 1 how this relates to Figure 1 - is this the expected birthrate of the red zone or of the purple and yellow zones. This needs moer explanation, and the sizes of the different Kebeles given to ensure proper balance and range of sizes. The buffer zones - and the fact there are not universally buffers between Kebeles in different arms needs to be explained as well as Figure 1 is introduced.

Response

Thank you for your insightful observations. We appreciate your concerns and would like to provide further clarification regarding Table 1 and Figure 1.

Firstly, Table 1 presents the expected number of pregnant mothers we anticipate evaluating from a specific kebele within a designated time period. This data is crucial for assessing our intervention outcomes. In contrast, Figure 1 illustrates the protection against information contamination by creating buffer zones, which was developed using ArcGIS.

Regarding the size differences among kebeles, our goal is to balance the total population between the intervention and control kebeles. It is important to note that achieving equal sizes for each kebele is not feasible, as two kebeles will inherently differ in size.

We acknowledge the need to explain the buffer zones and the fact that these zones may not be universally applicable between kebeles in different arms. Lastly, we confirm that there is a clear demarcation between kebeles, with each kebele having clear borders.

Reviewer comment

Please justify an ICC of 0.01 - where has this been taken from, and please demonstrate it is relevant to this setting.

Response

Thank you for your valuable suggestion. We have included the reference in the sample size determination section to clarify the justification for using an ICC of 0.01. This reference provides context and demonstrates its relevance to our specific setting.

Reviewer comment

We are told that 2000 pairs will be used here, but 150 per cluster - please discuss the discrepancy here.

Response

Thank you for your critical observation. You are correct that the total number of pairs needed is 2,000, translating to 1,000 pairs per arm. The figure of 150 refers to the expected average number of births in our study clusters. Specifically, the calculated average number of births during the study period was estimated to be 100 per cluster. However, this number may vary, potentially being higher or lower, depending on the total population within each kebele.

Reviewer comment

There is an issue with individual consent and cluster randomization in terms of differential consent by arm - one generally needs to explain how this is to be guarded against and what the expected consent rates are going to be - will willingness to consent increase the ICC? Is there any way of getting some data from routine records? Completion of visits is presumably routinely reported enabling an ITT analysis.The issue of precisely what data is to be collected and how analyzed is missing.

Response

Thank you for highlighting this important issue. We appreciate your concerns regarding individual consent and the potential for differential consent rates by arm, which could impact outcome assessments.To address this, we have incorporated the non-response rate into our sample size calculations and have planned for an intention-to-treat (ITT) analysis. Additionally, we aim to collect facility data related to antenatal care (ANC) and the timing of discharge after delivery for mothers who give birth at health facilities and have initiated ANC.We believe these measures will help mitigate the effects of non-response and ensure robust analysis throughout the study. Thank you for your understanding.

Kind regards.

---

## [Decision Letter · Decision Letter 2]

13 Feb 2025

Cluster randomized controlled trial to assess the effectiveness of a package of community-based intervention on continuum of maternal and newborn health care in Sidama, Ethiopia:The SiMaNeH trial protocol.

PONE-D-24-34008R2

Dear Dr. Tekle,

We’re pleased to inform you that your manuscript has been judged scientifically suitable for publication and will be formally accepted for publication once it meets all outstanding technical requirements.

Kind regards,

Anteneh Fikrie, MPH

Academic Editor

PLOS ONE

Additional Editor Comments (optional):

Reviewers' comments:

Reviewer's Responses to Questions

**Comments to the Author**

1. Does the manuscript provide a valid rationale for the proposed study, with clearly identified and justified research questions?

Reviewer #3: Yes

2. Is the protocol technically sound and planned in a manner that will lead to a meaningful outcome and allow testing the stated hypotheses?

Reviewer #3: Yes

3. Is the methodology feasible and described in sufficient detail to allow the work to be replicable?

Reviewer #3: Yes

4. Have the authors described where all data underlying the findings will be made available when the study is complete?

Reviewer #3: Yes

5. Is the manuscript presented in an intelligible fashion and written in standard English?

Reviewer #3: Yes

6. Review Comments to the Author

You may also provide optional suggestions and comments to authors that they might find helpful in planning their study.

Reviewer #3: Thank you for appropriately addressing my previous comments

7. PLOS authors have the option to publish the peer review history of their article (what does this mean? ). If published, this will include your full peer review and any attached files.

**Do you want your identity to be public for this peer review?** For information about this choice, including consent withdrawal, please see our Privacy Policy .

Reviewer #3: No

---

## [Editor Report · Acceptance letter]

PONE-D-24-34008R2

PLOS ONE

Dear Dr. Gebretsadik,

I'm pleased to inform you that your manuscript has been deemed suitable for publication in PLOS ONE. Congratulations! Your manuscript is now being handed over to our production team.

Kind regards,

on behalf of

Professor Anteneh Fikrie

Academic Editor

PLOS ONE